

# Effects of mental health interventions for students in higher education are sustainable over time: a systematic review and meta-analysis of randomized controlled trials

Regina Winzer[1,2,*], Lene Lindberg[1,3,*], Karin Guldbrandsson[1,2] and Anna Sidorchuk[1,4]

[1] Department of Public Health Sciences, Karolinska Institutet, Stockholm, Sweden
[2] Department of Living Conditions and Lifestyles, The Public Health Agency of Sweden, Solna, Sweden
[3] Center for Epidemiology and Community Medicine, Stockholm County Council, Stockholm, Sweden
[4] Department of Clinical Neuroscience, Centre for Psychiatry Research, Karolinska Institutet, Stockholm, Sweden

* These authors contributed equally to this work.

Corresponding author
Regina Winzer, regina.winzer@folkhalsomyndigheten.se

## ABSTRACT

**Background:** Symptoms of depression, anxiety, and distress are more common in undergraduates compared to age-matched peers. Mental ill health among students is associated with impaired academic achievement, worse occupational preparedness, and lower future occupational performance. Research on mental health promoting and mental ill health preventing interventions has shown promising short-term effects, though the sustainability of intervention benefits deserve closer attention. We aimed to identify, appraise and summarize existing data from randomized control trials (RCTs) reporting on whether the effects of mental health promoting and mental ill health preventing interventions were sustained at least three months post-intervention, and to analyze how the effects vary for different outcomes in relation to follow-up length. Further, we aimed to assess whether the effect sustainability varied by intervention type, study-level determinants and of participant characteristics.

**Material and Methods:** A systematic search in MEDLINE, PsycInfo, ERIC, and Scopus was performed for RCTs published in 1995–2015 reporting an assessment of mental ill health and positive mental health outcomes for, at least, three months of post-intervention follow-up. Random-effect modeling was utilized for quantitative synthesis of the existing evidence with standardized mean difference (Hedges' *g*) used to estimate an aggregated effect size. Sustainability of the effects of interventions was analyzed separately for 3–6 months, 7–12 months, and 13–18 months of post-intervention follow-up.

**Results:** About 26 studies were eligible after reviewing 6,571 citations. The pooled effects were mainly small, but significant for several categories of outcomes. Thus, for the combined mental ill health outcomes, symptom-reduction sustained up to 7–12 months post-intervention (standardized mean difference (Hedges' *g*) effect size (ES) = −0.28 (95% CI [−0.49, −0.08])). Further, sustainability of

symptom-reductions were evident for depression with intervention effect lasting up to 13–18 months (ES = −0.30 (95% CI [−0.51, −0.08])), for anxiety up to 7–12 months (ES = −0.27 (95% CI [−0.54, −0.01])), and for stress up to 3–6 months (ES = −0.30 (95% CI [−0.58, −0.03])). The effects of interventions to enhance positive mental health were sustained up to 3–6 months for the combined positive mental health outcomes (ES = 0.32 (95% CI [0.05, 0.59])). For enhanced active coping, sustainability up to 3–6 months was observed with a medium and significant effect (ES = 0.75 (95% CI [0.19, 1.30])).

**Discussion:** The evidence suggests long-term effect sustainability for mental ill health preventive interventions, especially for interventions to reduce the symptoms of depression and symptoms of anxiety. Interventions to promote positive mental health offer promising, but shorter-lasting effects. Future research should focus on mental health organizational interventions to examine their potential for students in tertiary education.

# INTRODUCTION

Mental health problems among students in higher education is an emerging public health issue and evidence-based prevention is essential (*Christensson et al., 2010*; *Dahlin et al., 2011*; *Garlow et al., 2008*; *Hunt & Eisenberg, 2010*; *Steptoe et al., 2007*). Recent systematic reviews on student health raise concerns over high rates of mental ill health outcomes with pooled prevalence ranging between 27% and 34% for depression and depressive symptoms and reaching 11% for suicidal ideation (*Ibrahim et al., 2013*; *Rotenstein et al., 2016*; *Tung et al., 2018*). Also, a two-fold risk for suicide is shown during ongoing university studies compared to when having attained university studies (*Lageborn et al., 2017*). Elevated rates of mental ill health, namely symptoms of distress, anxiety, and depression, in undergraduates appear to substantially exceed the corresponding estimates in age-matched peers (*Cvetkovski, Reavley & Jorm, 2012*; *Dyrbye, Thomas & Shanafelt, 2006*; *Leahy et al., 2010*; *Winzer et al., 2014*) and the general population (*Ibrahim et al., 2013*; *Rotenstein et al., 2016*). Female students, minority groups, and students with financial problems constitute groups with higher risks (*Cvetkovski, Reavley & Jorm, 2012*; *Eisenberg, Hunt & Speer, 2013*; *Said, Kypri & Bowman, 2013*). Once heightened at the beginning of the study period, the symptoms of anxiety, and depression remain elevated over the academic years and at no time point drop down to pre-registration levels (*Bewick et al., 2010*). Mental ill health among students may potentially be caused by heavy workload, insufficient feedback from teachers and worries about future endurance/competence (*Dahlin, Joneborg & Runeson, 2005*), but may also reflect the increase in deteriorated mental health among adolescents (*Hunt & Eisenberg, 2010*). Mental ill health problems are often accompanied by decrements in positive

mental health through lowered self-perception, inadequate social–emotional skills, and poor interpersonal relationships (*Conley, Durlak & Kirsch, 2015*). Moreover, perceived academic stress and burn-out are associated with impaired academic achievement (*Andrews & Wilding, 2004*; *Keyes et al., 2012*; *Vaez & Laflamme, 2008*), worse occupational preparedness and lower occupational performance after the graduation (*Rudman & Gustavsson, 2012*). In prevention science the health promotion approach constitutes a substantial ingredient of the integrative model for mental health intervention in youth (*Weisz et al., 2005*). Including aspects of positive mental health, i.e., emotional, psychological and social well-being (*Westerhof & Keyes, 2010*) is a beneficial strategy in mental health interventions (*Kobau et al., 2011*). It has been shown that psychological assets, e.g., boostering positive emotions, coping strategies, and compassion may help people to manage life's challenges. Thus, promoting mental health and preventing mental ill health are two essential and complementary steps in reducing the burden of disease (*Jane-Llopis, 2007*; *Keyes, 2007*; *World Health Organization (WHO), 2002*).

Previous research on mental health promotion and mental ill health prevention has shown promising short-term effects of stress reduction techniques and meditation, self-hypnosis, cognitive behavioral, and mindfulness interventions (*Conley, Durlak & Kirsch, 2015*; *Conley et al., 2017*; *Regehr, Glancy & Pitts, 2013*; *Shiralkar et al., 2013*) as well as of technology based interventions (*Conley et al., 2016*; *Davies, Morriss & Glazebrook, 2014*; *Farrer et al., 2013*). As mental health problems persist during the study period (*Bewick et al., 2010*; *Christensson et al., 2010*) and negatively affect academic performance and future working capacity (*Rudman & Gustavsson, 2012*; *Vaez & Laflamme, 2008*), the sustainability of intervention benefits as well as its determinants and moderators deserve closer attention. Several reviews have approached the issue of intervention effect sustainability by averaging the effects reported for the longest follow-up periods, although a substantial variability in the ranges and means of the follow-up lengths made the comparisons difficult (*Conley, Durlak & Kirsch, 2015*; *Conley et al., 2016*, *2017*). The authors highlighted the need for in-depth investigation of the intervention benefit sustainability over variable post-intervention follow-up periods since the effects may change their direction and strength over time (*Conley, Durlak & Kirsch, 2015*). Therefore, to further address the nature of sustainability of intervention effects, in this review we aimed to systematically identify, appraise and summarize the existing data from randomized control trials (RCTs) reporting on whether the effects of mental health promoting and mental ill health preventing interventions are sustained for at least three months of post-interventional follow-up. Further, we aimed to analyze how the direction and magnitude of the effects vary for different outcomes in relation to the lengths of follow-up and to assess whether effect sustainability varied by the types and major features of interventions, study-level determinants, and characteristics of participants.

## MATERIALS AND METHODS

### Eligibility criteria

The protocol was registered in PROSPERO, CRD42015029353 (Data S1). The study followed the guidelines for conducting systematic review as suggested by the Cochrane

handbook for systematic reviews of interventions (*Higgins, Green & Cochrane Collaboration, 2008*) and reported the study findings and procedure in relation to the sec statement (*Moher et al., 2009*) (Table S5).

The PICO components (population, intervention, comparator and outcome) were developed after discussing eligibility criteria with stakeholders from the student health services: P = students in university settings; I = any types of mental health-promoting and mental ill health- preventing interventions; C = any types of active or inactive controls; O = (i) positive mental health, including well-being, coping, locus of control, resilience, self-esteem/self-compassion, stress management, academic achievement or academic performance, and (ii) mental ill health, including symptoms of anxiety, symptoms of depression, psychological distress, worry, fatigue, sleeping problems, and perceived stress. The study design was restricted to RCTs with at least three months of post-intervention follow-up. No language restrictions were initially applied. Studies focused on students with diagnosed psychiatric disorders and studies conducted in primary care settings were excluded.

## Search strategy

In collaboration with librarians (CG, AW: see Acknowledgements), a sensitive search strategy was developed and adapted to the following databases: MEDLINE (Ovid), PsycInfo (Ovid), ERIC (Ovid), and Scopus. Gray literature was searched in Dissertations & Theses (ProQuest), Dart Europe, OpenGrey, and Base Bielefeld. The searches were limited to studies published from January 1, 1995 to December 31, 2015. Key words and MESH terms are reported in the Supplemental Information (Data S2 and Data S3, respectively). Reference lists of the relevant reviews and studies selected for inclusion were manually scrutinized and the following journals were hand-searched from January 2012 to March 2016: *College Student Journal, Journal of American College Health and Journal of College Counseling.*

## Study selection, data extraction, and quality assessment

Screening was conducted independently by two authors (RW, KG) and two colleagues (AF, AM: see Acknowledgements). The eligibility of each article was initially evaluated by the title and abstract and, if found appropriate, followed by full-text examination. At this stage only English-language publications were assessed. This resulted in a loss of 20 publications in Chinese ($k = 15$), Japanese ($k = 3$), Korean ($k = 1$), and Spanish ($k = 1$). Gray literature was taken into consideration when accessible free of charge. Any disagreements were resolved through panel discussions. Studies selected for inclusion were examined for potential overlap in study populations, which was not found.

Data extracted from the articles included first author, country of origin, setting, funding, inclusion and exclusion criteria, characteristics of the intervention and comparison groups (age, gender, ethnicity), characteristics of the intervention (type, format, delivery level, length of session, duration), type of comparison, outcome definition and measurement scale, sample size, post-intervention length of follow-up, percent of withdrawals at each measurement point, and study quality (described below).

If a study reported multiple outcomes and/or if outcomes were assessed at multiple follow-up time points, quantitative data (means and standard deviations (SDs)) were extracted separately for each outcome at each follow-up period. The same approach was utilized for multi-armed RTCs, from which separate extraction was performed for each intervention–comparison pair. When data were missing in the original reports we contacted authors for further clarification.

As suggested by *Conley, Durlak & Kirsch (2015)*, original interventions were grouped into: (i) cognitive behavior therapy (CBT)-related if focusing on identifying and changing unhelpful cognitions, behaviors and emotional regulation; (ii) mind–body-related, i.e., interventions that facilitate the mind's capacity to affect bodily function and symptoms; and (iii) psycho-educational-related if focusing on information, discussion and didactic communication on, e.g., stress-reduction and coping. Categorization was based on the original definitions, if provided, and otherwise by us. Level of delivery was considered as universal if intervention targeted students without reported mental ill health symptoms and as selective if provided for those with adverse mental health symptoms. Interventions were further divided into group or individual format. Comparators were sub-divided into active controls (i.e., another type of intervention) and inactive controls (waitlist controls, placebo-controls, "living as usual," and no intervention). Study outcomes were classified in two major categories: mental ill health outcomes consisting of anxiety symptoms, depressive symptoms, psychological distress, stress, self-reported worry, passive coping, and deteriorated quality of sleep; and positive mental health and academic performance outcomes including self-esteem, self-compassion, self-efficacy, mental or subjective well-being, resilience, active coping, happiness, stress management and academic performance.

The quality of selected trials was assessed independently by three authors (RW, KG, LL) and colleagues (AF, AM, SB: see Acknowledgements) using the Effective Public Health Practice Project Quality Assessment Tool (EPHPP), as recommended by the Cochrane Collaboration for public health reviews (*Higgins, Green & Cochrane Collaboration, 2008*). The EPHPP assesses selection bias, study-design, confounders, blinding, data collection method and withdrawals, and dropouts to yield the study quality as either strong, moderate or weak. Discrepancies in quality assessment were resolved by discussions with one of the three reviewers not involved in the review process.

## Statistical analysis

Because of the variety of instruments used for measuring outcomes, a standardized mean difference using Hedges' *g* was chosen as a common effect size (ES) for conducting quantitative synthesis. ES was calculated separately at each post-intervention follow-up time point as a difference in means between intervention and control group, divided by the pooled within-group SD and incorporating a correction factor for small sample sizes (*Borenstein et al., 2009*). One trial reported Hedges' *g* as the study ES (*Braithwaite & Fincham, 2009*), while for other studies it was calculated from the available raw data. Throughout the recalculations we kept the original direction of scales indicating the improvement of outcome measures. Thus, for mental ill health outcomes ESs below zero

pointed to superiority of the intervention group over the controls, while for positive mental health and academic performance outcomes, ESs above zero indicated that the results favored the intervention. For one study where follow-up means, but not SDs, were provided and the intervention effect was indicated as "non-significant" (*Chiauzzi et al., 2008*), we set ES to zero. To ease interpretation of the magnitude of Hedges' *g*, we applied Cohen's convention (*Cohen, 1992*) and defined the ES as small (0.2), medium (0.5), and large (0.8).

Precautions were taken to overcome unit-of-analysis error and avoid using multiple assessment of the same construct (*Higgins, Green & Cochrane Collaboration, 2008*). If more than one ES was reported in a given study for the same outcome at a given follow-up point (e.g., for depression assessed by both the Hamilton depression rating scale and Beck depression inventory), we averaged ESs to obtain the single outcome measure per intervention at each measurement point (*Higgins, 2006*; *Jones & Johnston, 2000*; *Kanji, White & Ernst, 2006*; *Peden et al., 2001*; *Seligman et al., 1999*). ESs were also averaged within the trials with multiple interventions of similar nature, i.e., if interventions belonged to the same category (*Chiauzzi et al., 2008*; *Mak et al., 2015*). A similar approach was applied to studies with multiple comparisons (*Chiauzzi et al., 2008*; *Rohde et al., 2014*; *Yang et al., 2014*). An exception was the study by *Kanji, White & Ernst (2006)*, where two control groups—an attention control and a time control—were included separately as considered to be different in approach and content and, thus, representing active and inactive comparisons, respectively.

Meta-analysis was conducted for all specific outcomes originally reported and for outcomes combined within mental ill health and positive mental health and academic performance categories. To analyze the combined outcomes, we applied a hierarchical approach for selecting outcomes from the studies reporting more than one from the same category. The hierarchy was based on descending order of outcome reporting, i.e., from the most often reported to the least often reported. For mental ill health outcomes the hierarchical selection was ordered as: depressive symptoms, anxiety symptoms, stress, psychological distress, self-reported worry, quality of sleep, and passive coping. For positive mental health and academic performance outcomes the order was: self-esteem, academic performance, self-efficacy, self-compassion, mental or subjective well-being, resilience, stress management, active coping, and happiness.

Because of the initial assumptions of between-study heterogeneity, a random-effects model incorporating both within- and between-study variability was used for quantitative synthesis. To assess the sustainability of intervention effect over time and address the variety of the follow-up lengths reported in the original studies, we categorized the post-intervention follow-ups as 3–6 months, 7–12 months, and 13–18 months. Each of the included studies reported outcome measures for at least one of these categories and quantitative synthesis was conducted separately for each category. If a given study provided several outcome measures falling in the same length category (e.g., for both three and six month follow-ups), the ES for the follow-up close to the upper boundary (i.e., six months) was chosen (*Kanji, White & Ernst, 2006*; *Seligman, Schulman & Tryon, 2007*; *Vazquez et al., 2012*). Only one study (*Seligman et al., 1999*) assessed outcomes at

follow-up periods longer than 18 months and the measurements of those periods (i.e., 24 months, 30 months, and 36 months) were not included in the meta-analysis. Finally, to obtain comparability between trials, if a study provided results based on both imputed data (i.e., intention-to-treat analysis) and non-imputed data (i.e., follow-up completers) (*Reavley et al., 2014*) or if both crude and adjusted ESs were available (*Chase et al., 2013*) we favored the imputed and crude measures for the main analysis leaving the latter (non-imputed and adjusted measures) for sensitivity analysis.

We evaluated statistical heterogeneity among the studies using $Q$ and $I^2$ statistics. For $Q$, $p$-value $< 0.1$ was considered as representative of statistically significant heterogeneity, and $I^2$ values of 25%, 50%, and 75% were indicating low, moderate and high heterogeneity, respectively (*Higgins et al., 2003*).

The subgroup analyses were performed by stratifying the main analysis by a priori identified moderators related to interventions (category of intervention, delivery level, type of format, type of controls), study-level moderators (initial study size, study quality) and moderators related to participant characteristics (gender, country). The analyses were performed if at least two studies were included in each subgroup. Mixed-methodology was applied with random-effect modeling used for within-group pooling, while between-group differences were assessed with fixed-effect model. Leave-one-out influence analysis was conducted to assess the potential impact of individual studies on the overall pooled ES by omitting one study at a time (*Tobias, 1999*). Following the approach suggested by *Hart et al. (2012)*, sensitivity analysis was conducted to assess whether the overall pooled ES differed if the lowest or the highest original ES was selected from the studies with multiple outcome assessments or multiple interventions or comparisons. In meta-analyses with three or more studies included, we assessed publication bias by funnel plots, Egger's regression asymmetry test, and the Begg–Mazumdar adjusted rank correlation test (*Begg & Mazumdar, 1994*; *Egger et al., 1997*).

All statistical analyses were performed using STATA version 13.1 (StataCorp, College Station, TX, USA), $p$-values $< 0.05$ were considered statistically significant, and all statistical tests were two-sided.

## RESULTS

After removing the duplicates, 6,571 records were available for title and abstract screening. Among these, 6,519 records were excluded as not meeting the PICO-criteria, leaving 52 articles for full-text examination. Further evaluation excluded another 26 studies: post-intervention follow-up less than three months ($k = 11$), not enough data to calculate ES ($k = 6$), not a RCT ($k = 5$), population not relevant ($k = 3$), and outcome not relevant ($k = 1$). A selection process yielded a final number of 26 RCTs to be included in the meta-analysis (Fig. 1).

### Characteristics of included studies

Table 1 summarizes the characteristics of studies eligible for inclusion. Among the 26 RCTs (*Braithwaite & Fincham, 2009*; *Chase et al., 2013*; *Cheng et al., 2015*; *Chiauzzi et al., 2008*;
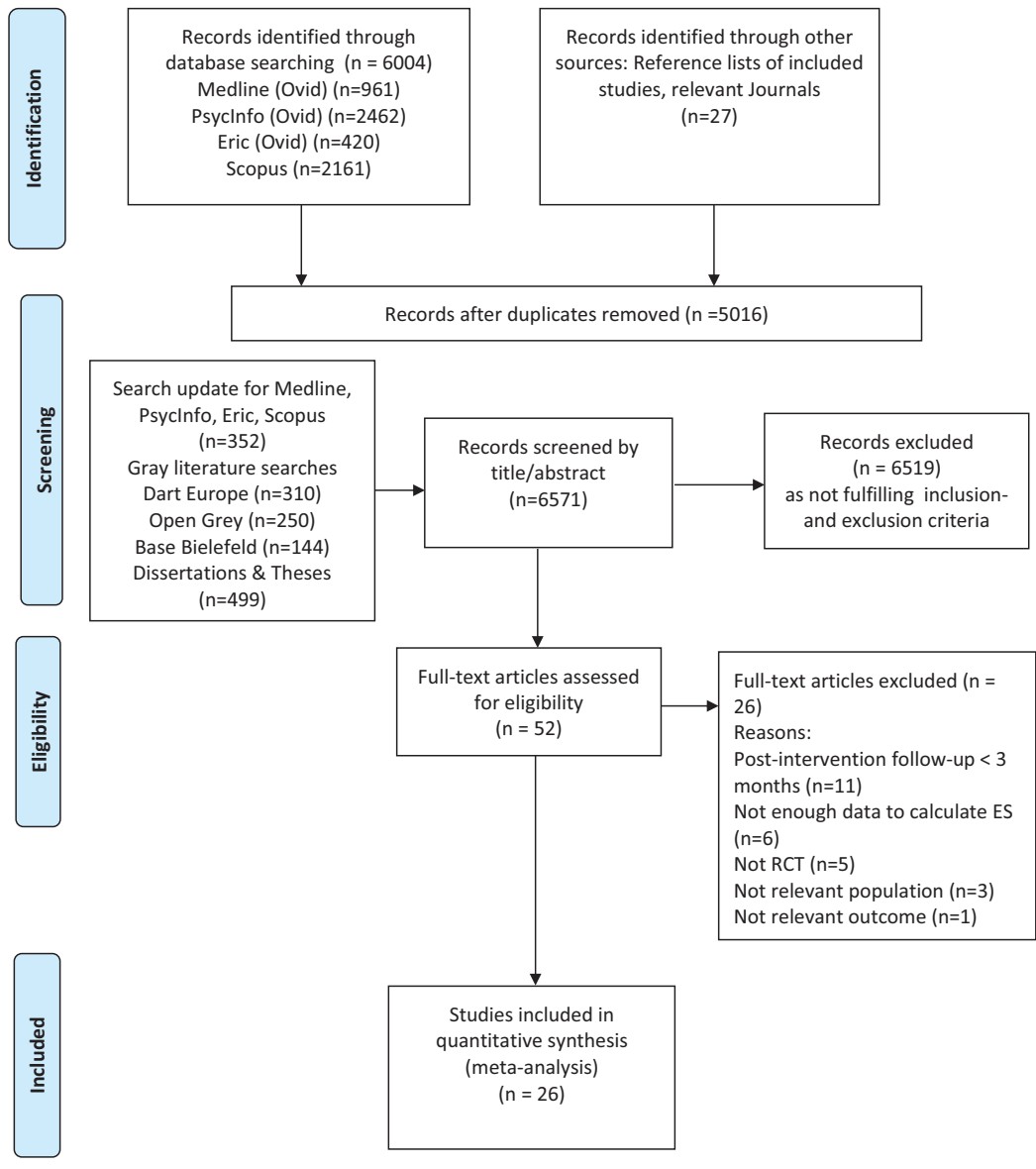

**Figure 1 Flow diagram of the study selection process.** PRISMA flow diagram (*Moher et al., 2009*).

*Erogul et al., 2014*; *Fontana et al., 1999*; *Franklin & Franklin, 2012*; *Gortner, Rude & Pennebaker, 2006*; *Hamdan-Mansour, Puskar & Bandak, 2009*; *Higgins, 2006*; *Jones & Johnston, 2000*; *Kanji, White & Ernst, 2006*; *Kattelmann et al., 2014*; *Kenardy, McCafferty & Rosa, 2006*; *Li et al., 2015*; *Mak et al., 2015*; *Pachankis & Goldfried, 2010*; *Peden et al., 2001*; *Reavley et al., 2014*; *Rohde et al., 2014*; *Seligman et al., 1999*; *Seligman, Schulman & Tryon, 2007*; *Shapiro et al., 2011*; *Vazquez et al., 2012*; *Yang et al., 2014*; *Zheng et al., 2015*), CBT-related interventions were assessed in 11 studies, while mind–body-related and psycho-educational-related interventions were assessed in 10 and five studies, respectively. Universal and selective delivery levels were equally present ($k = 13$ for both). Face-to-face group format was the most common ($k = 16$). At least one mental ill health outcome

**Table 1 Summary of study characteristics of randomized controlled trials included in the systematic review and meta-analysis.**

|  | Number of comparisons ($k$) | % |
|---|---|---|
| Total number of studies | 26 | 100 |
| **Participant characteristics** | | |
| Gender mix, study population | | |
| Approx. even (40%–60% females) | 6 | 23.0 |
| More than 60% females | 17 | 65.0 |
| More than 60% males | 1 | 4.0 |
| Not reported | 2 | 8.0 |
| Region (countries) | | |
| US | 14 | 54.0 |
| Australia | 3 | 11.5 |
| Europe (UK, Scotland, Spain) | 3 | 11.5 |
| East Asia and Pacific (China + Hong Kong) | 5 | 19.0 |
| Middle East and North Africa (Jordan) | 1 | 4.0 |
| **Intervention characteristics** | | |
| Intervention classification | | |
| CBT-related | 11 | 42.0 |
| Mind–body-related | 10 | 39.0 |
| Psycho-educational-related | 5 | 19.0 |
| Enlarged with material/home-work/training/booster | | |
| Yes | 15 | 58.0 |
| No/Unclear | 11 | 42.0 |
| Type of delivery | | |
| Universal | 13 | 50.0 |
| Selective | 13 | 50.0 |
| Type of format | | |
| Internet-based individual | 5 | 19.0 |
| Internet-based individual and in groups | 2 | 8.0 |
| Face-to-face individual | 2 | 8.0 |
| Face-to-face in group | 16 | 61.0 |
| Face-to-face in pairs | 1 | 4.0 |
| Length of intervention | | |
| <One week | 4 | 15.0 |
| One to four weeks | 1 | 4.0 |
| Five to seven weeks | 7 | 27.0 |
| Eight weeks | 8 | 31.0 |
| Nine to 12 weeks | 5 | 19.0 |
| 13–16 weeks | 0 | 0.0 |
| >16 weeks | 1 | 4.0 |

(Continued)

| | Number of comparisons ($k$) | % |
|---|---|---|
| **Table 1 (continued).** | | |
| **Comparison condition** | | |
| Active control[a] | 8 | 31.0 |
| Inactive control[b] | 18 | 69.0 |
| **Study characteristics** | | |
| Length of follow-up[c] | | |
| Three months | 13 | 50.0 |
| Four to six months | 14 | 54.0 |
| Seven to nine months | 3 | 11.5 |
| 10–12 months | 7 | 27.0 |
| 13–15 months | 0 | 0.0 |
| >15 months | 5 | 19.0 |
| Study size (participants) | | |
| $n \leq 100$ | 16 | 62.0 |
| $n > 100$ | 10 | 38.0 |
| Study quality[d] | | |
| Weak | 10 | 39.0 |
| Moderate | 12 | 46.0 |
| Strong | 4 | 15.0 |

Notes:
[a] For example, a different variant of the same intervention, a different intervention.
[b] For example, no intervention, "living as usual," a waiting list control.
[c] Percentage does not add to 100 because studies could fall into multiple categories.
[d] Assessed by The Effective Public Health Practice Project Quality Assessment Tool (EPHPP).

was assessed in 24 studies, while at least one positive mental health outcome was appraised in 14 studies. Twenty-three trials reported at least one outcome measurement during 3–6 months post-intervention follow-up, with eight and five trials reporting corresponding measurements during 7–12 months and 13–18 months follow-ups. None of the interventions had an organizational approach. More detailed characteristics of the selected studies are presented on-line (Table S1). The study quality assessed in all 26 RCTs varied between strong ($k = 4$), moderate ($k = 12$), and weak ($k = 10$). When subdivided by the outcome categories, 24 trials with at least one mental ill health outcome revealed their study quality as strong ($k = 4$), moderate ($k = 11$), and weak ($k = 9$), and 14 trials with at least one positive mental health outcome and academic performance of strong ($k = 3$), moderate ($k = 6$), and weak ($n = 5$) quality. Across all studies included in the analysis, selection bias was the most commonly assessed weakness component ($n = 21$). (Table S2).

## Effects and sustainability over time
### Interventions preventing mental ill health
As presented in Table 2, for the combined mental ill health outcomes an aggregated ES for all preventive interventions yielded a superiority of interventions over the comparisons at 3–6 months and 7–12 months of post-intervention follow-up, although

Table 2 Meta-analysis and sub-group analyses for hierarchically selected mental ill health outcomes, stratified by the length of post interventional follow-up periods.

| Variables | Length of post intervention follow-up periods (months) | | |
|---|---|---|---|
| | 3–6 | 7–12 | 13–18 |
| **All interventions ($k$)** | 21 | 9 | 3 |
| Hedges' $g$ (95% CI) | −0.28 (−0.44, −0.12) | −0.28 (−0.49, −0.08) | −0.17 (−0.39, 0.05) |
| $Q/I^2$ | 97.50***/79.5% | 31.00***/74.2% | 4.78[(*)]/58.2% |
| **Subgroup analyses** | | | |
| *Type of interventions* | | | |
| CBT-related ($k$) | 11 | 4 | 2 |
| Hedges' $g$ (95% CI) | −0.40 (−0.64, −0.16) | −0.12 (−0.51, 0.16) | −0.30 (−0.51, −0.08) |
| $Q/I^2$ | 44.60***/77.6% | 7.61[(*)]/60.6% | 0.28/0.0% |
| Mind–body-related ($k$) | 9 | 3 | 0 |
| Hedges' $g$ (95% CI) | −0.20 (−0.44, 0.04) | −0.43 (−0.66, −0.20) | – |
| $Q/I^2$ | 33.30***/76.0% | 1.00/0.0% | – |
| Psycho-educational ($k$) | 1 | 2 | 1 |
| Hedges' $g$ (95% CI) | 0.09 (−0.05, 0.23) | −0.64 (−1.83, 0.54) | −0.02 (−0.16, 0.16) |
| $Q/I^2$ | – | 16.58***/94.0% | – |
| Group difference $Q$ (d$f$)/$p$ for $Q$ | 3.38 (1)/0.06 | 5.82 (2)/0.05 | n/a |
| *Delivery level* | | | |
| Universal ($k$) | 8 | 5 | 1 |
| Hedges' $g$ (95% CI) | −0.23 (−0.46, −0.01) | −0.46 (−0.83, −0.09) | −0.02 (−0.16, 0.12) |
| $Q/I^2$ | 34.23***/79.6% | 23.04***/82.6% | – |
| Selective ($k$) | 13 | 4 | 2 |
| Hedges' $g$ (95% CI) | −0.31 (−0.54, −0.08) | −0.12 (−0.39, 0.16) | −0.30 (−0.51, −0.08) |
| $Q/I^2$ | 55.74***/78.5% | 7.61[(*)]/60.6% | 0.28/0.0% |
| Group difference $Q$ (d$f$)/$p$ for $Q$ | 7.48 (1)/0.006 | 0.35 (1)/0.55 | n/a |
| *Format type* | | | |
| Face-to-face in group ($k$) | 15 | 6 | 2 |
| Hedges' $g$ (95% CI) | −0.35 (−0.54, −0.16) | −0.20 (−0.42, 0.03) | −0.30 (−0.51, −0.08) |
| $Q/I^2$ | 55.20***/74.6% | 11.68*/57.2% | 0.28/0.0% |
| Face-to-face individual ($k$) | 2 | 0 | 0 |
| Hedges' $g$ (95% CI) | 0.06 (−0.30, 0.42) | – | – |
| $Q/I^2$ | 1.91/47.5% | – | – |
| Internet-based individual ($k$) | 2 | 2 | 0 |
| Hedges' $g$ (95% CI) | −0.28 (−1.09, 0.53) | −0.64 (−1.83, 0.54) | – |
| $Q/I^2$ | 6.26*/84.0% | 16.58***/94.0% | – |
| Internet-based individual and in groups ($k$) | 2 | 1 | 1 |
| Hedges' $g$ (95% CI) | −0.30 (−1.10, 0.50) | −0.45 (−0.84, −0.07) | −0.02 (−0.16, 0.12) |
| $Q/I^2$ | 15.17***/93.4% | – | – |
| Group difference $Q$ (d$f$)/$p$ for $Q$ | n/a | 0.56 (1)/0.45 | n/a |

(Continued)

| Variables | Length of post intervention follow-up periods (months) | | |
|---|---|---|---|
| | **3–6** | **7–12** | **13–18** |
| *Type of comparison* | | | |
| Active (*k*) | 2 | 2 | 0 |
|   Hedges' *g* (95% CI) | −0.34 (−1.21, 0.53) | −0.88 (−1.59, −0.18) | – |
|   $Q/I^2$ | 12.40***/91.9% | 4.45*/77.5% | – |
| Inactive (*k*) | 19 | 7 | 3 |
|   Hedges' *g* (95% CI) | −0.28 (−0.44, −0.11) | −0.14 (−0.28, 0.01) | −0.17 (−0.39, 0.05) |
|   $Q/I^2$ | 83.38***/78.4% | 10.17/41.0% | 4.78/58.2% |
| Group difference *Q* (d*f*)/*p* for *Q* | 1.67 (1)/0.20 | 16.39 (1)/<0.001 | n/a |
| *Study quality* | | | |
| Strong (*k*) | 2 | 2 | 1 |
|   Hedges' *g* (95% CI) | −0.22 (−0.63, 0.19) | −0.17 (−0.34, 0.01) | −0.26 (−0.52, −0.01) |
|   $Q/I^2$ | 2.50/60.0% | 0.06/0.0% | – |
| Moderate (*k*) | 10 | 4 | 0 |
|   Hedges' *g* (95% CI) | −0.26 (−0.44, −0.09) | −0.38 (−0.99, 0.24) | – |
|   $Q/I^2$ | 21.12*/57.4% | 21.82***/86.2% | – |
| Weak (*k*) | 9 | 3 | 2 |
|   Hedges' *g* (95% CI) | −0.30 (−0.62, 0.02) | −0.23 (−0.61, 0.02) | −0.15 (−0.50, 0.20) |
|   $Q/I^2$ | 71.20***/88.8% | 8.29*/75.9% | 2.82/64.6% |
| Group difference *Q* (d*f*)/*p* for *Q* | 2.63 (2)/0.27 | 0.85 (2)/0.65 | n/a |
| *Study size* | | | |
| 100 participants or less (*k*) | 14 | 7 | 1 |
|   Hedges' *g* (95% CI) | −0.45 (−0.68, −0.23) | −0.38 (−0.71, −0.05) | −0.39 (−0.80, 0.02) |
|   $Q/I^2$ | 44.08***/70.5% | 23.91**/74.9% | – |
| More than 100 participants (*k*) | 7 | 2 | 2 |
|   Hedges' *g* (95% CI) | −0.01 (−0.10, 0.08) | −0.09 (−0.20, 0.00) | −0.12 (−0.34, 0.12) |
|   $Q/I^2$ | 7.07/15.1% | 0.69/0.0% | 2.66/62.4% |
| Group difference *Q* (d*f*)/*p* for *Q* | 46.3 (1)/<0.001 | 6.41 (1)/0.01 | n/a |
| *Participants' gender mix* | | | |
| Approx. even (40%–60% females) (*k*) | 3 | 1 | 1 |
|   Hedges' *g* (95% CI) | −0.31 (−0.65, 0.04) | −0.16 (−0.34, 0.02) | −0.26 (−0.52, −0.01) |
|   $Q/I^2$ | 5.40/63.0% | – | – |
| More than 60% females (*k*) | 14 | 8 | 2 |
|   Hedges' *g* (95% CI) | −0.32 (−0.52, −0.11) | −0.32 (−0.58, −0.06) | −0.15 (−0.50, 0.20) |
|   $Q/I^2$ | 80.27***/83.3% | 31.0***/77.4% | 2.82/64.6% |
| More than 60% males (*k*) | 1 | 0 | 0 |
|   Hedges' *g* (95% CI) | 0.30 (−0.17, 0.77) | – | – |
|   $Q/I^2$ | – | – | – |
| Not reported (*k*) | 3 | 0 | 0 |
|   Hedges' *g* (95% CI) | −0.25 (−0.94, 0.44) | – | – |
|   $Q/I^2$ | 7.73*/74.1% | – | – |

| Variables | Length of post intervention follow-up periods (months) | | |
|---|---|---|---|
| | **3–6** | **7–12** | **13–18** |
| Group difference $Q$ (d$f$)/$p$ for $Q$ | 0.45 (2)/0.80 | n/a | n/a |
| *Country* | | | |
| US ($k$) | 10 | 6 | 2 |
|   Hedges' $g$ (95% CI) | −0.22 (−0.47, 0.03) | −0.20 (−0.45, 0.06) | −0.30 (−0.51, −0.08) |
|   $Q/I^2$ | 35.73***/74.8% | 21.48**/76.7% | 0.28/0.0% |
| Other countries ($k$) | 11 | 3 | 1 |
|   Hedges' $g$ (95% CI) | −0.33 (−0.55, −0.11) | −0.45 (−0.66, −0.25) | −0.02 (−0.16, 0.12) |
|   $Q/I^2$ | 60.57***/83.5% | 0.71/0.0% | – |
| Group difference $Q$ (d$f$)/$p$ for $Q$ | 1.15 (1)/0.28 | 8.82 (1)/0.003 | n/a |

**Notes:**
The format type "face-to-face in pair" was not utilized for mental ill health outcomes.
$k$, number of studies; n/a, not applicable.
(*) $p < 0.1$.
* $p < 0.05$.
** $p < 0.01$.
*** $p < 0.001$.

the effects were small. Pooled ES did not reach statistical significance for follow-up periods of 13–18 months. High-to-moderate heterogeneity was detected at all three follow-up periods with $I^2$ of 79.5%, 74.2%, and 58.2%, respectively. Publication bias was evident for studies with 3–6 months follow-up (Egger's test $p$-value = 0.013), though not for studies with longer follow-up periods (7–12 months: $p$-value = 0.151; 13–18 months: $p$-value = 0.141), (Fig. S1). Influence analysis revealed no indication that individual RCTs, if omitted, would significantly influence the observed overall ESs. As previously noted, sensitivity analyses were performed for studies with multiple outcome measures reported for the same follow-up and for studies with multiple interventions or comparisons. Pooling together the highest ESs originally reported for these studies did not alter the results of the main analysis 3–6 months: ES = −0.28 (95% CI [−0.44, −0.12]); 7–12 months: ES = −0.35 (95% CI [−0.60, −0.10]); 13–18 months: ES = −0.23 (95% CI [−0.54, 0.08]). Neither were the results influenced when the lowest originally reported ESs were used (3–6 months: ES = −0.24 (95% CI [−0.39, −0.09]); 7–12 months: ES = −0.23 (95% CI [−0.43, −0.03]); 13–18 months: ES = −0.06 (95% CI [−0.18, 0.06]). Only one study reported the results using both imputed and non-imputed data (*Reavley et al., 2014*), with the former included in the main meta-analysis. Alternative inclusion of the latter did not change the overall ES.

In sub-group analyses for the combined mental ill health outcomes, studies employing CBT-related interventions revealed significant pooled ESs for 3–6 month and 13–18 month follow-ups (Table 2; Fig. 2). Less consistent results were observed for mind-body-related interventions. No superiority of intervention group appeared among studies with psycho-educational interventions. Pooled ESs for universal preventive interventions yielded significant results for follow-up up to 7–12 months. Less consistency appeared in the aggregated results for selective interventions and interventions conducted

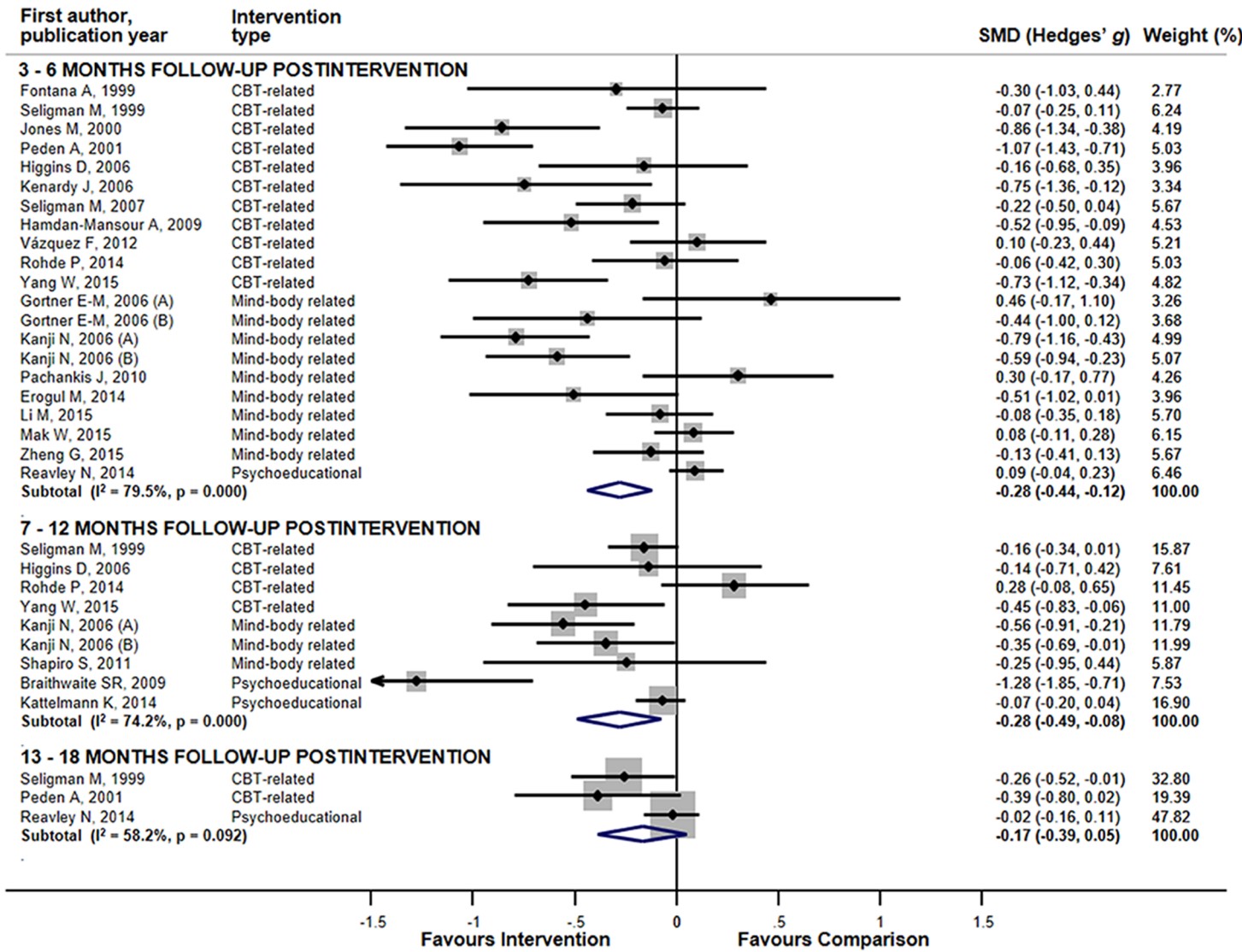

**Figure 2** The effects of mental ill health preventing interventions on hierarchically selected mental ill health outcomes stratified by the length of post-interventional follow-up periods. Lines represent standardized difference in means (Hedges' g) and 95% confidence intervals (CI); the size of the box represents the weight of each study.

face-to-face in groups. Trials with small sample size and trials comprising more than 60% females yielded significant effects for up to 7–12 months of follow-up. The small numbers of studies might explain the lack of consistency in the results of other sub-group comparisons. The high heterogeneity seen for studies with 3–6 months of follow-up might reflect differences in delivery level (p-value for Q between sub-groups = 0.006) and study size ($p < 0.001$), while for studies with follow-up periods of 7–12 months, heterogeneity could be explained by differences in type of comparison ($p < 0.001$), study size ($p = 0.01$), and country where the RCT was conducted ($p = 0.003$). We were unable to detect between-group differences for trials with follow-up of 13–18 months because of the small number of studies in the sub-groups.

Assessment of the specific mental ill health outcomes, revealed a sustainable effect of all interventions combined lasting up to 13–18 months for symptoms of depression (ES = −0.30 (95% CI [−0.51, −0.08])), (Table S3). For symptoms of anxiety sustainability was observed up to 7–12 months (ES = −0.27 (95% CI [−0.54, −0.01])). Only one study assessed the effect of interventions targeting anxiety during 13–18 months of post-intervention follow-up and, hence, we were unable to perform meta-analysis. For symptoms of stress, reductions lasted up to 3–6 months post-intervention (ES = −0.30 (95% CI [−0.58, −0.03])). Other comparisons were either inconclusive or quantitative synthesis was not performed because of the small number of studies.

### Interventions promoting mental health and academic performance— a paucity of outcomes

Table 3 presents overall ESs for the combined positive mental health and academic performance outcomes with rather limited data available, in particular, for the follow-up periods longer than 3–6 months. All interventions combined showed superiority over the controls during 3–6 months of follow-up with small, but significant pooled ES. For longer follow-up periods the results were inconclusive. High heterogeneity was detected when studies with 3–6 months follow-up were pooled ($I^2$ = 86.5%). Because of the small number of studies, publication bias were only assessed for studies with 3–6 months of follow-up and were detected (Egger's test $p$-value = 0.03) (Fig. S1). Influence analysis indicated that four individual studies (*Erogul et al., 2014*; *Hamdan-Mansour, Puskar & Bandak, 2009*; *Pachankis & Goldfried, 2010*; *Peden et al., 2001*), if omitted, would drop the significant overall ES for the studies with 3–6 months follow-up to borderline significance. Overall ESs at follow-ups of 7–12 and 13–18 months remain non-significant regardless of individual study influences. Sensitivity analyses pooling the highest ESs originally reported for studies with multiple outcome assessment or multiple interventions or comparison groups showed no alteration to the overall results at 3–6 months follow-up (ES = 0.32 (95% CI [0.06, 0.58])), but made the overall ESs for 7–12 months follow-ups significant (ES = 0.53 (95% CI [0.20, 0.87])), as well as for 13–18 months follow-up (ES = 0.53 (95% CI [0.21, 0.86])). However, only two studies were assessed within each category of 7–12 and 13–18 months follow-ups. Use of the lowest originally reported ES did not affect the results of the main analysis (3–6 months: ES = 0.32 (95% CI [0.06, 0.58]); 7–12 months: ES = 0.16 (95% CI [−0.18, 0.50]); 13–18 months: ES = 0.16 (95% CI [−0.17, 0.49])). One study reported both crude and adjusted outcome assessment (*Chase et al., 2013*). Use of the adjusted ES for sensitivity analysis did not alter the results observed in the main analysis.

Sub-group analyses for the combined positive mental health and academic performance outcomes were performed only for studies with 3–6 months follow-up (Table 3; Fig. 3). Superiority of interventions over comparisons was shown for CBT-related interventions, selective delivery level, face-to-face group format, RCTs with inactive comparisons, studies with small sample size and trials conducted in US. Between-group difference was significant for delivery level ($p < 0.001$), format type ($p < 0.001$), study size ($p < 0.001$), and gender mix ($p = 0.02$). Sub-group analyses for

**Table 3 Meta-analysis and sub-group analyses for hierarchically selected positive mental health and academic performance outcomes stratified by the length of post interventional follow-up periods.**

| Variables | Length of post intervention follow-up periods (months) | | |
|---|---|---|---|
| | **3–6** | **7–12** | **13–18** |
| **All interventions ($k$)** | 11 | 2 | 2 |
| Hedges' $g$ (95% CI) | 0.32 (0.05, 0.59) | 0.34 (−0.05, 0.73) | 0.33 (−0.06, 0.72) |
| $Q/I^2$ | 73.8***/86.5% | 0.00/0.0% | 0.04/0.0% |
| **Subgroup analyses** | | | |
| *Type of interventions* | | | |
| CBT-related ($k$) | 4 | 0 | 1 |
| Hedges' $g$ (95% CI) | 0.52 (0.06, 0.98) | – | 0.29 (−0.29, 0.87) |
| $Q/I^2$ | 24.5***/87.8% | – | – |
| Mind-body related ($k$) | 6 | 1 | 0 |
| Hedges' $g$ (95% CI) | 0.23 (−0.16, 0.61) | 0.35 (−0.35, 1.05) | – |
| $Q/I^2$ | 41.68***/88.0% | – | – |
| Psycho-educational ($k$) | 1 | 1 | 1 |
| Hedges' $g$ (95% CI) | 0.10 (−0.38, 0.58) | 0.34 (−0.14, 0.82) | 0.37 (−0.17, 0.91) |
| $Q/I^2$ | – | – | – |
| Group difference $Q$ (d$f$)/$p$ for $Q$ | | n/a | n/a |
| *Delivery level* | | | |
| Universal ($k$) | 5 | 2 | 1 |
| Hedges' $g$ (95% CI) | 0.05 (−0.23, 0.33) | 0.34 (−0.05, 0.74) | 0.37 (−0.17, 0.91) |
| $Q/I^2$ | 21.75**/77.0% | 0.00/0.0% | – |
| Selective ($k$) | 5 | 0 | 1 |
| Hedges' $g$ (95% CI) | 0.64 (0.18, 1.09) | – | 0.29 (−0.29, 0.87) |
| $Q/I^2$ | 34.8***/88.5% | – | – |
| Group difference $Q$ (d$f$)/$p$ for $Q$ | 17.26 (1)/<0.001 | n/a | n/a |
| *Format type* | | | |
| Face-to-face in groups ($k$) | 6 | 1 | 1 |
| Hedges' $g$ (95% CI) | 0.53 (0.16, 0.91) | 0.35 (−0.35, 1.05) | 0.29 (−0.29, 0.87) |
| $Q/I^2$ | 27.34***/81.7% | – | – |
| Face-to-face individual ($k$) | | | |
| $k$ | 2 | 0 | 0 |
| Hedges' $g$ (95% CI) | 0.51 (−0.63, 1.66) | – | – |
| $Q/I^2$ | 16.17***/93.8% | – | – |
| Face-to-face in pairs ($k$) | 0 | 1 | 1 |
| Hedges' $g$ (95% CI) | – | 0.34 (−0.14, 0.82) | 0.37 (−0.17, 0.91) |
| $Q/I^2$ | – | – | – |
| Internet-based individual ($k$) | 3 | 0 | 0 |
| Hedge's g (95% CI) | −0.14 (−0.36, 0.07) | – | – |
| $Q/I^2$ | 4.42/54.8% | – | – |
| Group difference $Q$ (d$f$)/$p$ for $Q$ | 25.88 (2)/<0.001 | n/a | n/a |

(Continued)

| Variables | Length of post intervention follow-up periods (months) | | |
|---|---|---|---|
| | 3–6 | 7–12 | 13–18 |
| *Type of comparison* | | | |
| Active (*k*) | 1 | 0 | 0 |
|   Hedges' *g* (95% CI) | −0.11 (−0.50, 0.28) | – | – |
|   $Q/I^2$ | – | – | – |
| Inactive (*k*) | 10 | 2 | 2 |
|   Hedges' *g* (95% CI) | 0.36 (0.08, 0.65) | 0.34 (−0.05, 0.74) | 0.33 (−0.06, 0.73) |
|   $Q/I^2$ | 72.53***/87.6% | 0.00/0.0% | 0.04/0.0% |
| Group difference *Q* (d*f*)/*p* for *Q* | n/a | n/a | n/a |
| *Study quality* | | | |
| Strong (*k*) | 2 | 1 | 0 |
|   Hedges' *g* (95% CI) | 0.49 (−0.53, 1.51) | 0.35 (−0.35, 1.05) | – |
|   $Q/I^2$ | 12.33***/91.9% | – | – |
| Moderate (*k*) | 5 | 0 | 0 |
|   Hedges' *g* (95% CI) | 0.18 (−0.11, 0.47) | – | – |
|   $Q/I^2$ | 17.02**/76.5% | – | – |
| Weak (*k*) | 4 | 1 | 2 |
|   Hedges' *g* (95% CI) | 0.48 (−0.30, 1.25) | 0.34 (−0.14, 0.82) | 0.33 (−0.06, 0.73) |
|   $Q/I^2$ | 43.94***/93.2% | – | 0.04/0.0% |
| Group difference *Q* (d*f*)/*p* for *Q* | 0.53 (2)/0.77 | n/a | n/a |
| *Study size* | | | |
| 100 participants or less (*k*) | 5 | 1 | 1 |
|   Hedges' *g* (95% CI) | 0.84 (0.46, 1.23) | 0.35 (−0.35, 1.05) | 0.29 (−0.29, 0.87) |
|   $Q/I^2$ | 11.97*/66.6% | – | – |
| More than 100 participants (*k*) | 6 | 1 | 1 |
|   Hedges' *g* (95% CI) | −0.04 (−0.20, 0.13) | 0.34 (−0.14, 0.82) | 0.37 (−0.017, 0.91) |
|   $Q/I^2$ | 12.01*/58.4% | – | – |
| Group difference *Q* (d*f*)/*p* for *Q* | 49.84 (1)/<0.001 | n/a | n/a |
| *Participants' gender mix* | | | |
| Approx. even (40%–60% females) (*k*) | 4 | 1 | 1 |
|   Hedges' *g* (95% CI) | 0.49 (−0.06, 1.03) | 0.34 (−0.14, 0.82) | 0.37 (−0.017, 0.91) |
|   $Q/I^2$ | 22.88***/86.9% | – | – |
| More than 60% females (*k*) | 6 | 1 | 1 |
|   Hedge's *g* (95% CI) | 0.10 (−0.19, 0.40) | 0.35 (−0.35, 1.05) | 0.29 (−0.29, 0.87) |
|   $Q/I^2$ | 29.73***/83.2% | – | – |
| More than 60% males (*k*) | 1 | 0 | 0 |
|   Hedges' *g* (95% CI) | 1.12 (0.62, 1.63) | – | – |
|   $Q/I^2$ | – | – | – |
| Not reported (*k*) | 0 | 0 | 0 |
|   Hedges' *g* (95% CI) | – | – | – |
|   $Q/I^2$ | – | – | – |

(Continued)

| Variables | Length of post intervention follow-up periods (months) | | |
|---|---|---|---|
| | 3–6 | 7–12 | 13–18 |
| Group difference $Q$ (d$f$)/$p$ for $Q$ | 5.18 (1)/0.02 | n/a | n/a |
| *Country* | | | |
| US ($k$) | 6 | 1 | 1 |
|   Hedges' $g$ (95% CI) | 0.52 (0.11, 0.93) | 0.35 (−0.35, 1.05) | 0.29 (−0.29, 0.87) |
|   $Q/I^2$ | 38.79***/87.1% | – | – |
| Other countries ($k$) | 5 | 1 | 1 |
|   Hedges' $g$ (95% CI) | 0.09 (−0.25, 0.44) | 0.34 (−0.14, 0.82) | 0.37 (−0.017, 0.91) |
|   $Q/I^2$ | 24.24***/83.5% | – | – |
| Group difference $Q$ (d$f$)/$p$ for $Q$ | 10.78 (1)/0.001 | n/a | n/a |

Notes:
The format type "Internet-based individual and in groups" was not utilized for positive mental health and academic performance outcomes.
$k$, number of studies; n/a, not applicable.
* $p < 0.05$.
** $p < 0.01$.
*** $p < 0.001$.

studies with longer follow-up revealed either non-significant results or were impossible to conduct owing to the small number of trials in the sub-groups.

Because of lack of data on the specific positive mental health and academic performance outcomes, only studies on active coping, self-esteem, and self-efficacy with 3–6 months follow-up were quantitatively assessed (Table S4). Sustainability of the intervention effect was observed for active coping (ES = 0.75 (95% CI [0.19, 1.30])) with no significant effects shown for other outcomes.

## DISCUSSION

Our systematic review and meta-analysis showed sustainability of the benefits of mental health interventions targeting students in higher education, though in most of the analyses, the pooled ESs yielded significant, but small overall effects. For the combined mental ill health outcomes, the observed effects across all preventive interventions were sustained for up to 7–12 months post-intervention. Sustainability of effects was most pronounced for interventions designed to reduce the symptoms of depression, for which the superiority of intervention groups over the comparisons remained significant for up to 13–18 months post-intervention. For the combined positive mental health and academic performance outcomes, aggregated results across all promotion interventions revealed slightly shorter, but still evident sustained effects, which remained significant at post-intervention follow-up of 3–6 months.

To our knowledge, this is the first systematic review and meta-analysis focusing primarily on the sustainability of the effects of mental health promoting and mental ill health preventing interventions among students in higher education and analyzing different categories of follow-up duration. A direct comparison to the existing literature was therefore difficult as other reviews mostly assessed the effects measured at the

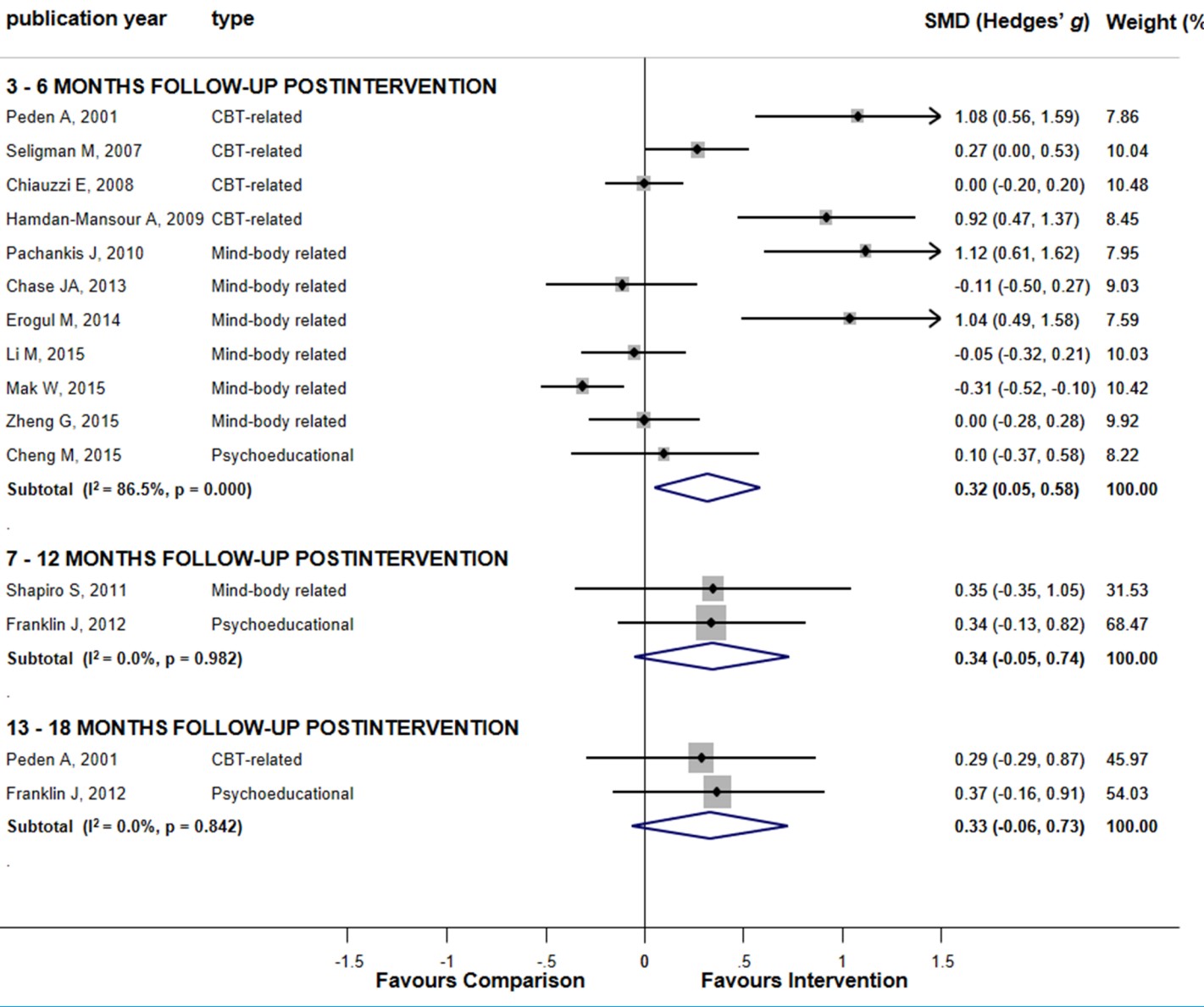

**Figure 3** **The effects of mental health promoting interventions on hierarchically selected positive mental health and academic performance outcomes stratified by the length of post-interventional follow-up periods.** The effect sizes of all interventions and combined subtotals for positive mental health and academic performance outcomes by the length of follow-up. Lines represent standardized difference in means (Hedges' *g*) and 95% confidence intervals (CI); the size of the box represents the weight of each study.

completion of interventions. The closest comparisons are three reviews by Conley et al. on universal and indicated mental health prevention programs (*Conley, Durlak & Kirsch, 2015*; *Conley et al., 2017*) and technology-delivered preventive interventions (*Conley et al., 2016*). The reviews by Conley et al., assessed the effects of interventions across all types of adjustment outcomes in university students at the longest follow-up period reported, which varied from two to 52 weeks (*Conley, Durlak & Kirsch, 2015*), 13 to 52 weeks (*Conley et al., 2016*), and four to 157 weeks (*Conley et al., 2017*). The first review

(*Conley, Durlak & Kirsch, 2015*) showed the duration of follow-up to be negatively correlated with aggregated ES across mental ill health and positive mental health outcomes combined as well as no effect for psycho-educational interventions. A similar tendency for the effects of intervention to become non-significant as the duration of follow-up increases was observed in our study, though the sustainability of effects differed between ill-health and positive mental health outcomes. As in Conley's review (*Conley, Durlak & Kirsch, 2015*), no effects of psycho-educational interventions on any outcomes were evident in our data, regardless of the duration of follow-up. The second review (*Conley et al., 2016*) reported a significant effect of universal interventions at any follow-up periods ranging between 13 and 52 weeks as well as a positive effect of selective interventions during the follow-up periods of 2–26 weeks. Similarly, in our study mental ill health outcomes were reduced by universal interventions for up to 7–12 months of follow-up and by selective interventions at follow-ups of up to 3–6 months, although our results on positive mental health and academic performance outcomes were less conclusive. Similar to the third review (*Conley et al., 2017*), our results indicated that the most sustainable effects were observed for interventions designed to reduce the symptoms of depression and symptoms of anxiety.

Although our literature search for intervention studies was not limited to psychological interventions, only this type was retrieved. The scarcity of organizational mental health promoting interventions was verified by a scoping review (*Enns et al., 2016*). However, an exception may be a recent systematic review on learning environment interventions for medical student well-being, suggesting changes to curriculum (*Wasson et al., 2016*). Their results support previous findings suggesting that to maximize the effectiveness of mental health promotion, all levels of delivery must contribute, i.e., not just individual and group levels, but also structural, and societal levels (*Hamilton & Bhatti, 1996*). To further improve the sustainability of student mental health promotion, psychological interventions may be combined with a whole-setting approach, as endorsed by the WHO initiative health promoting universities (HPU) (*World Health Organization, 1995*).

## Limitations

Systematic reviews on student mental health have indicated lack of follow-up data on outcome assessment as a major obstacle for determining the long-term effect of interventions (*Conley, Durlak & Kirsch, 2015*; *Conley et al., 2016*; *Davies, Morriss & Glazebrook, 2014*; *Farrer et al., 2013*). Likewise, the scarcity of studies assessing the effects of interventions at post-interventional follow-ups of longer than three months along with a substantial variability in the lengths of follow-ups reported in the original studies should be considered as major limitations of our review. In particularly, the lack of original evidence affected our analysis of positive mental health outcomes as it restricted us to mainly aggregating the effects of interventions with 3–6 months of follow-up. Other limitations must also be considered. First, in most cases low numbers of studies in sub-groups prevented us from exploring the moderating effect of types of interventions, study-level determinants and participant characteristics during follow-up

periods longer than six months making the results of sub-group analyses tentative. This also precluded us from conducting in-depth investigation of sources for heterogeneity, which was found to be mostly high. Second, our intention to analyse two dimensions of mental health, that resulted in combining the original outcomes into the "mental ill health" and "positive mental health and academic performance" outcome categories, with a hierarchical approach applied, could have boosted heterogeneity. In a subsequent analyses, we attempted to reduce heterogeneity by pooling together the studies with the same specific outcomes reported, though for several outcomes it was not possible due to data scarcity. Third, the evidence was insufficient to obtain any aggregated ESs for the specific outcomes, in particular, for self-reported worries, passive coping, academic performance, self-compassion, mental and subjective well-being, resilience and happiness rating. Fourth, a substantial variability exists in measurement instruments and, in several cases, the same outcome was measured by different scales. We tried to address this limitation by choosing Hedges' *g* as an ES and by investigating how sensitive the aggregated results were to our initial approach of combining the original ESs in cases of multiple outcome measures or in multi-armed RCTs. The sensitivity analyses proved the robustness of our findings for mental ill health, though for positive mental health outcomes the use of the lowest ESs from the original studies altered the results for studies with 7–12 and 13–18 months of follow-up. Fifth, more than 30% of the original studies were assessed as being of weak quality. To address this issue, we conducted sub-group analyses stratifying the trials by study quality. For both categories of outcomes, these analyses revealed inconclusive results when trials with insufficient quality were pooled that should be accounted when interpreting our results. Furthermore, selection bias was the most commonly identified weakness. This bias, whether induced by the investigators or caused by self-selection may have resulted in either underestimation or overestimation of the original ES and therefore could affect the aggregated results. Finally, the results should be seen in the context of the presence of publication bias among the studies with 3–6 months of follow-up and of our inability to assess publication bias for positive mental health outcomes at follow-ups longer than six months, which may have resulted from our restriction to English-language publications at the final stage of selection.

## CONCLUSION

Despite the limitations, the evidence suggests long-term effect sustainability for mental ill health preventive interventions, in particular, for interventions to reduce the symptoms of depression and symptoms of anxiety. Interventions designed to promote positive mental health offer promising, but shorter-lasting effects. As the research field of health promoting interventions for students expands, future studies may improve our attempts to establish the effectiveness and sustainability of those interventions, e.g., ascertaining the effects for specific positive mental health outcomes. In addition, future research should also focus on mental health organizational interventions to investigate their potential for students in tertiary education.

## ACKNOWLEDGEMENTS

We gratefully acknowledge Anders Wändahl and Carl Gornitzki, University Library, Karolinska Institutet for advice and support and for providing us with electronic searches; our colleagues, Annika Frykholm, Anna Månsdotter, and Sven Bremberg who contributed to the screening and extraction of data-base searches; and stake-holders from Student Health Services for valuable input for practice.

### Funding

The authors received no funding for this work.

### Competing Interests

The authors declare that they have no competing interests.

### Author Contributions

- Regina Winzer conceived and designed the experiments, performed the experiments, analyzed the data, contributed reagents/materials/analysis tools, authored or reviewed drafts of the paper, approved the final draft.
- Lene Lindberg conceived and designed the experiments, performed the experiments, analyzed the data, authored or reviewed drafts of the paper, approved the final draft.
- Karin Guldbrandsson performed the experiments, analyzed the data, authored or reviewed drafts of the paper, approved the final draft.
- Anna Sidorchuk performed the experiments, analyzed the data, contributed reagents/materials/analysis tools, prepared figures and/or tables, authored or reviewed drafts of the paper, approved the final draft.

### Supplemental Information

Supplemental information for this article can be found online at http://dx.doi.org/10.7717/peerj.4598#supplemental-information.

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
