# Peer review of "Effects of mental health interventions for students in higher education are sustainable over time: a systematic review and meta-analysis of randomized controlled trials"

_PeerJ, doi:10.7717/peerj.4598_

## Round 0.1 · original submission · Major Revisions

Thank you for your submission. The reviewers have stated a number of required changes that should be addressed, in particular with regard to your background, methodology and discussion sections. We look forward to receiving a revision version of your paper in due course.

Reviewer 1 ·

Basic reporting

In general the language is clear and professional throughout. However, I have some comments and suggestions for improvements:
- In the abstract, change the heading Conclusion to Discussion.
- Line 58: To state that a thing is common or has high prevalence without specifying how common or high is annoying.
- Line 66: Is it really higher risk groups and not groups with higher risk?
- Merge sentences lines 92-93
- Check heading of Table 2 and Figure 3.
- The layout of Tables could be improved by not using marked lines for every row and column. Also the authors could consider using background coloring of e.g. every other row.

Introduction
While the introduction gives the context to mental ill health – literature being well referenced and relevant – no corresponding context is given to positive mental health. There is no theoretical conceptualization of positive mental health, which would give the rationale for including (or not) certain outcomes as measures of positive mental health. Just because there are records of studies in the literature and methods to make effects of intervention sustainability across studies “statistically comparable” using standardized units, it doesn´t necessarily make them meaningful to include in a meta-analysis. Thus, if the authors want to keep positive mental health they should provide the relevant context and conceptualization that justify the outcomes included in the analyses and why they think they are meaningful to pool. If they do this my guess is that it will become evident that the scope of the study is better limited to mental ill health and related endpoints meaningful in a meta-analysis.

Experimental design

The research is original and within the aims and scope of the journal. However, the research question (s) could be better defined. It is unclear what is meant by mental health and mental ill health, for example. Also I do not like the part of the aim that addresses the implementation quality. While implementation quality is (!) important and should be brought to attention I do not think this is the way to do it. The reason is that RCT:s usually do not report implementation components which could lead to a bias in an assessment of such quality. Further, it could be stated more clearly how the present systematic review and meta-analysis differs from the previous ones (line 87). The authors refer to three previous reviews (Conley et al 2015, 2016, 2017) and it would have been easy to describe each of them more in detail. The present description makes it difficult to know what specifically their review adds. Since the referred reviews of Conley et al are recently published this becomes really important because it is the part where the novelty of the study should be stated clearly.

The methods are described with sufficient detail and information to replicate. An exception though, is found on line 106. Since there are multiple guidelines the authors should specify which were used.

Validity of the findings

The study deals with an important topic and it is evident that the authors have done an ambitions job trying to evaluate intervention sustainability of over time. That being said there are several limitations with the study, some of which are addressed by the authors in the discussion and some that I would like to stress further or add even. I have a major concern about collapsing so many types of outcomes into a single concept particularly for positive mental health. The studies included in the review or meta-analysis should be as homogeneous as possible. Now they introduce a potential bias that can be large. The high-to-moderate heterogeneity is a consequence of the broad outcome categories and an important limitation of the study. Collapsing outcomes also lead to other problems, like having to decide on a hierarchical order for positive outcomes - that in my mind do not make sense. Is academic performance a better indicator of positive mental health than well-being, for example? Effect sizes have to be excluded when studies report in more than one domain, which is negative because they fulfill the inclusion criteria but the effect size for another outcome is used.

In the fourth limitation of the study it is mentioned that 30% of the original studies were assessed as being of weak quality with selection bias as the most commonly identified weakness and what may have come out of this limitation. It would have been interesting to have the study quality distributed among the two major outcome categories, ill-health and positive. This would have helped the reader to evaluate the effects. A description of what type of quality impairments that was most common for each outcome concept and would also provide a larger picture. Lastly, I am skeptical about including the studies of weak quality in the meta-analysis.

Reviewer 2 ·

Basic reporting

(1) The paper is well written and easy to follow. The language throughout the paper is mostly clear and unambiguous. There are a few small suggestions for clarification, which I have included in the sections below as they seem more appropriate for clarifying issues around design and validity of findings rather than issues with grammar or writing style.

(2) The authors have provided a substantial and entirely relevant background for issues of student mental health, which leads well to the need for a study such as this.

(3) Except for a few small issues with formatting of the paper (e.g., full-stops in headings, table & figure headers, table formatting), the paper is formatted consistently. These formatting issues should be addressed, but they are minor.

(4) Raw data has been shared.

Experimental design

(1) This research is certainly original in its aims, well designed, and within scope of PeerJ.

(2) The research aims are provided and they give good structure to the paper overall. However, some (simple, but important) attention is needed to clarify these research aims.
- In the abstract, it is not clear whether the second last sentence (lines 27-29) is referring to the first aim, or describing a second aim, or prefacing the aim in the last sentence. Some small adjustments to the sentence will clarify this confusion, eg: if it is prefacing the aim in the last sentence, then the full-stop in line 27 could be replaced by a comma.
- A new (third?) aim is presented in lines 101-102, but is not mentioned in your abstract. The section in the results that addresses this aim is lacking detail: almost as though it is a secondary and less important element of the paper. I suggest either including this aim in the abstract (and providing more detail in the relevant results section) or removing this aim from the paper.
- Line 97: I recommend providing an example of what you mean by "too little information is often provided on the process of program delivery" - eg, what sort of information is generally missing? how would including that information benefit the evaluation or outcome? Clarifying this will help strengthen the argument for your method and identifying the gap in the literature that you are addressing.

(4) The research approach does very well to navigate the often-unclearly-defined areas of student mental health and mental ill-health. This is particularly noted in the comprehensive and in-depth method section, which I found easy to follow and in sufficient detail to allow for replication.
Two small points of clarification would further strengthen the description of the study's methodology:
- Line 42: It is not clear what you mean by "individual outcomes" in this sentence, as "individual outcomes" suggests outcomes for individual people, but the rest of the sentence describes outcomes for specific symptoms across the dataset. Perhaps a more accurate phrase than "individual outcomes is "specific symptoms of mental illness"?
- Also on line 42-43, the phrase "most considerable sustainability of symptoms reduction was evident" is awkwardly phrased. Consider adding "the" to the beginning of the phrase (ie "the most considerable..") and replacing "symptoms reduction was" with either "symptom-reduction was" or "symptom-reductions were".
One point of clarification would strengthen the results section:
- Lines 354-355 presents a substantial problem with the structure of the results and setup of (one part of) the papers' methodology (this problem can be easily resolved): if there is a "lack of data on individual positive mental health and academic performance outcomes", then why is this section heading "interventions promoting mental health and academic performance"? I recommend changing the heading to better represent the analysis taking place, and either: justify why you have up to this point focused your argument on positive mental health and academic performance without then analysing those outcomes, or, change the focus of your setup in the introduction to better account for an analytical focus on experiences such as active coping, self-esteem, and self-efficacy.

Validity of the findings

(1) The effect sizes reported in this paper are small (except for the effect size for active coping). This is fine, but rather than claiming the effect sizes are "substantial", I recommend that the authors acknowledge throughout their paper that the effect sizes are "small but significant" or "small but meaningful", as appropriate. This will prevent any misinterpretation of the actual patterns reported in the paper by novice / uninitiated readers.

(2) The data were shared with PeerJ. Data were handled in appropriate and reasonable ways. Findings are statistically sound, with the exception of my first point in this section.

(3) The concluding paragraph reflects the general findings in a realistic and thoughtful manner. The sentence addressing policy-makers and student health providers (lines 467-368) is not particularly helpful, and might be better focused on the different contexts and group differences (eg, medical students, group differences for intervention types, etc).
- The conclusion in the abstract is not as helpful as the overall conclusion. For example, Lines 52-54: "Interventions targeting university students … should be considered by student health policy-makers and providers": Considering that you have not investigated the attention paid to interventions by policy makers (and considering that many student health policymakers and providers are already invested in or searching for effective interventions), this concluding statement is not particularly helpful or meaningful for readers. It also does not do justice to the importance of your findings (that interventions have a small but real sustained effect!!) or relate back to your aims (investigating how sustainability varies for different types of interventions, in different contexts, with different cohorts, etc). I recommend considering a stronger conclusion in this abstract that speaks to the potential benefits of specific intervention types, and how these might vary in different contexts.

Additional comments

Well done on conducting such an innovative and important study! I was thoroughly interested and engaged in your paper, and I look forward to seeing it in print. I hope that you find my comments and suggestions useful for refining an already high-quality paper. My suggestions are all minor and (should you agree with them) they will not take much work to address.

---

## Round 0.2 · accepted · Accept

Thank you for your re-submission. I have reviewed this version and find that you have addressed all of the issues raised by the reviewers to my satisfaction and am therefore pleased to recommend this paper for publication

#